# Gut Microbiota and Liver Transplantation: Immune Mechanisms behind the Rejection

**DOI:** 10.3390/biomedicines11071792

**Published:** 2023-06-23

**Authors:** Ludovico Abenavoli, Giuseppe Guido Maria Scarlata, Maria Rosaria Paravati, Luigi Boccuto, Francesco Luzza, Emidio Scarpellini

**Affiliations:** 1Department of Health Sciences, University “Magna Graecia”, 88100 Catanzaro, Italy; l.abenavoli@unicz.it (L.A.); giuseppeguidomaria.scarlata@unicz.it (G.G.M.S.); mrparavati@unicz.it (M.R.P.); luzza@unicz.it (F.L.); 2School of Nursing, Healthcare Genetics Program, Clemson University, Clemson, SC 29634, USA; lboccut@clemson.edu; 3School of Health Research, Clemson University, Clemson, SC 29634, USA; 4Translationeel Onderzoek van Gastro-Enterologische Aandoeningen (TARGID.), Gasthuisberg University Hospital, KU Leuven, Herestraat 49, 3000 Leuven, Belgium

**Keywords:** liver transplantation, gut microbiota, liver cirrhosis, immunity, dysbiosis

## Abstract

Liver transplantation (LT) is the treatment of choice for patients with cirrhosis, decompensated disease, acute liver failure, and hepatocellular carcinoma (HCC). In 3–25% of cases, an alarming problem is acute and chronic cellular rejection after LT, and this event can lead to the need for new transplantation or the death of the patient. On the other hand, gut microbiota is involved in several mechanisms sustaining the model of the “gut–liver axis”. These include modulation of the immune response, which is altered in case of gut dysbiosis, possibly resulting in acute graft rejection. Some studies have evaluated the composition of the gut microbiota in cirrhotic patients before and after LT, but few of them have assessed its impact on liver rejection. This review underlines the changes in gut microbiota composition before and after liver transplantation, hypothesizing possible immune mechanisms linking dysbiosis to transplantation rejection. Evaluation of changes in the gut microbiota composition in these patients is therefore essential in order to monitor the success of LT and eventually adopt appropriate preventive measures.

## 1. Introduction

Liver transplantation (LT) is the treatment of choice for patients with cirrhosis, decompensated disease, acute liver failure, and hepatocellular carcinoma (HCC) who meet certain acceptance criteria, such as those of Milan, Pittsburgh, Toronto, Hangzhou, and the University of California, San Francisco [1,2]. Despite this, there are new therapies aimed at remodeling the liver parenchyma, reducing bile accumulation in the liver, or acting as antivirals in Hepatitis B Virus (HBV) and Hepatitis C Virus (HCV)-related cirrhosis [3,4,5,6,7]. In the last case, the occurrence of mutations in the viral genome may result in therapeutic failure, preventing the achievement of sustained virologic response (SVR) [8,9]. In 2021, the volume of liver transplantations continued to grow, with a record 9234 transplantations performed in the United States; 8665 (93.8%) from deceased donors and 569 (6.2%) from living donors [10]. Postoperative complications leading to LT failure are known. These are classified as (i) vascular (stenosis and thrombosis of the hepatic artery, portal vein, and inferior vena cava; pseudoaneurysm of the hepatic artery; arteriovenous fistula; and celiac stenosis) (ii) and biliary (stenosis, bile leakage, obstruction, recurrent disease, and infection) [11]. These complications affect patient survival at one, five, and ten years post-transplantation, known to be around 87.43%, 73.83%, and 61.23%, respectively, in males and 86.28%, 74.19%, and 65.10%, respectively, in females (*p* = 0.05) [12]. In 3–25% of cases, an alarming problem is acute and chronic cellular rejection after LT, mainly due to the action of recipient T cells recognizing donor alloantigens, causing a cytopathic immune response [13]. The use of tacrolimus has decreased the incidence of chronic cellular rejection, but, despite this, its occurrence can lead to the need for new transplantation or the death of the patient [14]. On the other hand, the intestinal microbiota is involved in several mechanisms involving the gut–liver axis. These include modulation of the immune response, which is altered in the case of gut dysbiosis [15]. Microbial dysbiosis is able to affect the transplantation organ’s tolerance or rejection [16]. A crucial step is the activation of Pattern Recognition Receptors (PRRs) present in immune and parenchymal cells of the liver, following exposure of these cells to translocated microbiotas’ bacteria. PRRs are present on both the membrane and cytosol of these cells and include several subfamilies of receptors that trigger different signaling pathways. The activation of various PRRs represents the strongest stimulus for modulating the immune response. Depending on the specific PRR activated, the microbiota and its metabolites affect the immune system differently [17,18]. The present review underlines the changes in gut microbiota composition before and after liver transplantation, hypothesizing possible immune mechanisms linking dysbiosis to transplantation rejection.

## 2. Materials and Methods

### Literature Review

A narrative review of the literature was performed through PubMed, NCBI, and Scopus search engines. Mesh terms were the keywords: “liver transplantation” and “gut microbiota”, “immunity”, and “dysbiosis”. The search included English papers published in each period. All types of papers were included, i.e., reviews, retrospective analyses, and experimental studies (Figure 1) [19].

## 3. Gut Microbiota Changes before and after Liver Transplantation

The gut microbiota is an ecosystem composed of more than 35,000 bacterial species that perform several functions including gut barrier protection, immunomodulation, and metabolic [20]. Bacterial colonization occurs as early as birth, and its composition tends to normalize within the first 2–3 years of life, though it is influenced by factors such as environmental, genetic, and nutritional [21]. The dominant gut microbial *phyla* are *Firmicutes*, *Bacteroidetes*, *Actinobacteria*, *Proteobacteria*, *Fusobacteria*, and *Verrucomicrobia*, with the two *phyla Firmicutes* and *Bacteroidetes* representing 90% of gut microbiota [22]. The *phylum Bacteroidetes* consists mainly of the genus *Bacteroides*, which is also often isolated from clinical specimens and is responsible for various pathological conditions [23]. On the other side, the *phylum Firmicutes* is represented by a large number of genera, including Gram-positive and Gram-negative bacteria [24]. The microbiota colonizes the gut–liver axis and is responsible for certain biochemical pathways that are not yet fully elucidated. This reciprocal interaction is established by the portal vein, which allows the transport of gut-derived products directly to the liver, and by the hepatic feedback pathway of biliary and antibody secretion to the gut [25]. For this reason, the role of the gut microbiota and the implications of its dysbiosis in liver diseases are currently under study [26,27,28,29].

### 3.1. Gut Microbiota and Dysbiosis Related to Prognosis on the Waiting List

Few studies have evaluated changes in the composition of the gut microbiota in the period before LT. Grat M. et al. collected stool samples from 40 patients with liver cirrhosis (LC) listed for the first LT [30]. Each fecal sample was seeded on appropriate selective and differential culture media and then incubated under the appropriate growth conditions to perform bacterial counts. Additionally, the microbial load was related to certain clinical and laboratory parameters. As reported by the authors, *Bifidobacterium* and *Enterococcus* species are mainly involved in gut dysbiosis among liver transplantation candidates. Specifically, the pre-transplantation dysbiosis ratio (PTDR) was significantly correlated with *Enterococcus* (R = −0.897; *p* < 0.001) but not with *Bifidobacterium* (R = 0.098; *p* = 0.546) counts. *Bifidobacterium* (standardized regression coefficient [sβ] = −0.549; *p* < 0.001), *Enterococcus* (sβ = 0.369; *p* = 0.004), and yeast (sβ = 0.315; *p* = 0.018) numbers were independently associated with serum bilirubin, while *Escherichia coli* counts (sβ = 0.318; *p* = 0.046) was correlated with the international normalized ratio (INR), and *Bifidobacterium* counts (sβ = 0.410; *p* = 0.009) with serum creatinine. Only *Bifidobacterium* (sβ = −0.468; *p* = 0.003) and *Enterococcus* (sβ = 0.331; *p* = 0.029) counts were independent predictors of the model for end-stage liver disease (MELD) score. In conclusion, the *Bifidobacterium/Enterococcus* ratio, proposed as a measure of pre-LT gut dysbiosis, was significantly related to the MELD score following the adjustment for the absolute *Bifidobacterium* (sβ = −0.333; *p* = 0.029) and *Enterococcus* (sβ = −0.966; *p* = 0.003) number. This study was carried out only at a single time point. Thus, it is necessary to expand the study to the follow-up of patients in order to confirm the association between the MELD score and *Bifidobacterium/Enterococcus* ratio.

An observational study conducted by Vangara S. et al. showed differences in gut microbiota composition among LT candidates and healthy donors [31]. Twenty-eight stool samples were collected between the two study groups for next-generation sequencing (NGS) of ribosomal RNA (rRNA) V3-V4 variable regions. *Bacteroidetes* were higher in the donor group than in the LT candidate group (48.3% vs. 31.9%, *p* = 0.007), while *Proteobacteria* (7.71% vs. 26.57%, *p* = 0.001) were reduced. Regarding the genera, LT candidates had a reduced abundance of *Prevotella* (8.9% vs. 29.4%; *p* < 0.05), *Lachnospira* (4.16% vs. 10.1%; *p* < 0.05), but a higher abundance of *Enterobacteriaceae* (20.1% vs. 3.5% *p* < 0.05), compared to healthy donors. The association between microbiota and prognostic scores showed that *Proteobacteria* and *Enterobacteriaceae* were positively correlated, and *Prevotella* and *Lachnospira* were negatively correlated, with increasing MELD scores (*p* < 0.05). Finally, events associated with portal hypertension were associated with increased *Actinobacteria*, *Bacteroidetes*/*Firmicutes* ratio, and decreased *Lactobacillaceae*, *Bifidobacteria,* and *Ruminococcus* (*p* < 0.05). Although this study needs to be expanded on a significantly larger cohort of patients, it is evident that dysbiosis was correlated with increased MELD scores and with a larger incidence of portal hypertension events, which may negatively influence prognosis.

### 3.2. Gut Microbiota and Dysbiosis after Liver Transplantation

Several interesting studies have evaluated changes in the gut microbiota composition between pre- and post-LT. For example, Wu ZW et al. conducted a four-year observational study in China to evaluate the changes in gut microbiota composition and immune parameters in LT recipients [32]. Stool and blood samples were collected from 190 participants, stratified as follows: 28 healthy subjects, 51 cirrhotic patients, and 111 liver-transplanted patients. Real-time quantitative polymerase chain reaction (RT-qPCR) and enzyme-linked immunosorbent assay (ELISA) were used to evaluate gut microbiota composition and immune parameters, respectively. The *Eubacteria*, *Bifidobacterium* spp., *Faecalibacterium prausnitzii,* and *Lactobacillus* spp. number of copies was significantly lower in the LT group than in healthy control subjects. Oppositely, the *Enterobacteriaceae* and *Enterococcus* spp. number of copies was significantly higher (*p* < 0.05). Comparison between the cirrhotic patient group and those undergoing LT showed no statistically significant differences in gut microbiota composition. Of mention, among the liver-transplanted patient group, *Faecalibacterium prausnitzii* was significantly lower in liver cancer than in liver cirrhosis patients. Furthermore, a gut microbiota analysis was performed at different time points within the liver-transplanted patient group, showing that *Bifidobacterium* spp., *Faecalibacterium prausnitzii*, *Lactobacillus* spp., *Bifidobacterium* spp., and *Enterobacteriaceae* returned to normal levels upon 13–24 months after LT. On the other hand, *Enterococcus* spp. abundance stayed significantly higher after LT, even after approximately two years (*p* < 0.05). Regarding correlation with laboratory parameters, authors showed a significant increase in plasma endotoxin, interleukin-6 (IL-6), and Secretory Immunoglobulin A (SIgA) in the cirrhotic patients’ group only. In the latter, plasma endotoxin and IL-6 levels were negatively correlated with all *Eubacteria* and the *Bacteroides–Prevotella* group abundance. The authors concluded the investigation, suggesting the importance of extending the evaluation of the gut microbiota composition of patients undergoing LT even after several years of major surgery. This is relevant to assess the effectiveness of any treatment for gut dysbiosis.

Sun LY et al. collected 33 stool samples from 9 patients with end-stage liver disease (one sample for the pre- and post-LT period, for a total of 18 samples) and 15 healthy controls to evaluate the microbial composition by sequencing of V4 16S rRNA [33]. The relative abundances of *Actinobacillus*, *Escherichia*, *Shigella*, *Fusobacteriales*, *Clostridium*, and *Aeromonas* were significantly decreased. Conversely, *Desulfobacterales*, *Sarcina*, and *Akkermansia* were significantly increased in post-LT samples vs. with pre-LT samples. However, there were no differences in the relative abundances between the post-LT and control groups. Furthermore, metabolic pathway analysis showed that M00116, a member of a metabolic pathway for the synthesis of vitamin K, was higher in post-LT patients. At the same time, M00232/M00233, an amino acid transporter in a metabolic system, was increased in patients after LT, indicating that the level of intestinal nutrients is increased and the digestive and absorptive functions of the intestinal tract are improved. Despite this study being conducted on a small sample size, it is interesting how the gut microbiota composition of post-LT patients was similar to those of healthy controls.

Bajaj JS et al. enrolled 68 cirrhotic outpatients listed for LT in a time span of five years to evaluate the gut microbiota composition with stool samples using 16s rRNA (V1-V2) sequencing [34]. Patients were divided into three different groups: pre-LT, post-LT, and controls. Alpha diversity between pre-LT and post-LT groups showed a significative increase measured using the Chao1 index (pre-LT 468.9 ± 263.1 vs. post-LT 694.9 ± 240.6, *p* < 0.05). At *phylum* levels, there were no significative changes in relative abundance (*Firmicutes* pre 37% vs. 53% post, *p* = 0.83, *Bacteroidetes* pre 53% vs. 37% post, *p* = 0.61, *Proteobacteria* pre 1.3 vs. 1.6% post, *p* = 0.65) However, at the genus level, there was a reduction in the relative abundance of potentially pathogenic bacteria (e.g., *Escherichia*, *Salmonella*, *Shigella*) and an increase in potentially beneficial bacteria (namely, *Ruminococcus* and *Lachnospira*). In comparison with the control group, it was recorded the highest diversity (Chao1 772.1 ± 254.2, *p* < 0.05) among pre- and post-LT groups. Furthermore, the control group showed a higher relative abundance for *Ruminococcus* and *Lachnospira* vs. the post-LT group. The latter showed an increase in *Bacteroides* vs. the control group. Overall, this study highlights an improvement in gut microbiota composition, with a higher relative abundance of “good” bacteria and a reduction in potentially pathogenic bacteria, within 6 months of LT. However, there is “residual” dysbiosis between post-LT patients and healthy controls.

The same research team enrolled another 40 cirrhotic patients on the LT list and followed them until 6 months post-LT to evaluate the gut microbiota diversity according to stool and metabolomic and proteomic profiles derived from urine and serum samples, respectively [35]. The results regarding gut microbiota composition were similar to the previous study. Interestingly, patients showed a reduction in serum endotoxin and a significative increase in total bile acid profile (*p* < 0.05) after LT. Regarding urinary metabolomics, there were differences in urinary nuclear magnetic resonance (NMR) spectroscopy as a whole, pre- and post-LT. These included a significant increase in trimethylamine-N-oxide (TMAO) (increased 2.4 folds, *p* = 0.01) and in phenylacetylglutamine (PAG) (increased 2.7 folds, *p* = 0.02) after LT. On the other hand, serum lipidomics showed significant changes after LT: those across several lipid classes, diacylglycerols, cholesterol esters, lyso-phosphethonolamine/phosphoethanolamine, and sphingomyelin (*p* < 0.05). Finally, there was a significant reduction in venous blood ammonia concentration (*p* = 0.001). This study reinforced previous findings sustaining the concept that LT leads to a slow recovery of eubiosis conditions that are closely related to endotoxin synthesis, ammonia, bile acid modulation, and methylamine metabolism. This underlines the central role of the gut–liver axis in metabolic processes in this subset of patients.

Going deeper into the influence of gut dysbiosis on immune functioning in LT patients, Lee SK et al. evaluated the relationship between these issues in long-term post-LT patients [36]. In particular, twenty-seven post-LT patients were enrolled and stratified as follows: 22 long-term post-LT group (under immunosuppressants) and 5 tolerant patients. Blood and fecal microbiota analysis were performed using flow cytometry and 16S rRNA sequencing, respectively. Gut microbiota analysis showed that *Faecalibacterium prausnitzii* was significantly decreased (*p* = 0.0032), and the *Bifidobacterium longum* and *Bifidobacterium bifidum* were marginally decreased in the long-term post-LT patients vs. the control subjects. To verify the immunomodulatory function of these microbes, long-term post-LT patients were treated with *Faecalibacterium prausnitzii*, *Bifidobacterium longum*, *Bifidobacterium bifidum*, and *Akkermansia muciniphila*. Of importance, regulatory T (Treg) cells significantly increased and T-helper 17 (Th17) cells were significantly decreased after treatment with *Faecalibacterium prausnitzii* (*p* = 0.0022)*, Bifidobacterium longum* (*p* = 0.0099), *Bifidobacterium bifidum* (*p* = 0.0006) and *Akkermansia muciniphila* (*p* = 0.0016). In further detail, in tolerant patients, *Faecalibacterium prausnitzii* was marginally increased (*p* = 0.053), coupled with an increase in Treg cells (*p* < 0.001), vs. the long-term post-LT group. Overall, the reduction in microbial diversity represented by *Faecalibacterium prausnitzii* showed a consensual decrease in Treg cells in long-term post-LT patients compared to healthy subjects. For all this evidence, the evaluation of the immunomodulatory activity of *Faecalibacterium prausnitzii* could be used as a potential biomarker for assessing the immune status and as a target for improving immune homeostasis in long-term post-LT patients.

A recent study conducted by Lai Z. et al. evaluated changes in gut microbiota composition in 37 adult Chinese patients at different time points: before LT (BLT; 16 samples), one week after LT (LT1W; 16 samples), two weeks after LT (LT2W; 16 samples) [37]. At the same time, a control group (CG; 21 samples) was included in the study. Gut microbiota analysis from stool samples was performed with 16S rRNA V3-V4 sequencing. The α-diversity using the Shannon index showed a significant reduction in richness in the LT1W and LT2W groups vs. the other groups (*p* < 0.05). In parallel, Unifrac β-diversity showed a significant diversity between CG and BLT groups, while the BLT group showed a significant diversity within the LT1W and LT2W groups. There was no significant diversity between LT1W and LT2W groups. At the *phylum* level, *Firmicutes* and *Bacteroidetes* were abundant in each study group: 57.50% and 37.379% (CG), 48.83% and 40.61% (BLT), 48.018% and 25.59% (LT1W), 37.16% and 26.07% (LT2W), respectively. At the genus level, the gut microbiota of the CG and BLT groups contained high levels of *Bacteroides* and *Faecalibacterium*. In contrast, the LT1W and LT2W groups were characterized by a higher content of *Bacteroides*, *Enterococcus*, *Escherichia-Shigella*, *Bifidobacterium*, *Lactobacillus,* and *Erysipelatoclostridium*. However, limitations of the study include a low sample size and low sequencing depth. Indeed, the study provides for the first time a “signature” of the microbiota characteristic of the Chinese population, highlighting microbial diversity pre- and postoperatively.

### 3.3. Gut Microbiota and Dysbiosis Related to Graft Rejection

Few studies have evaluated changes in the gut microbiota composition in acute graft rejection. Kato K. et al. retrospectively analyzed the gut microbiota of 38 patients who underwent LT at Kyoto University Hospital [38]. The V3-V4 16S rRNA sequencing with stool samples collected at different times (2 weeks before LT and every 7 days after LT during the first two months) was performed. Furthermore, gut composition after acute graft rejection or bloodstream infection (BSI) was evaluated vs. patients not experiencing these events. The administered immunosuppressants included tacrolimus, mycophenolate mofetil, and low-dose steroids started within 24 h of LT in all patients. *Firmicutes* (56.6%), *Bacteroidetes* (21.2%), and *Actinobacteria* (14.4%) were the most dominant *phyla* before LT. However, throughout LT, the mean diversity index decreased and then gradually increased during our observation period. *Proteobacteria* and *Actinobacteria* increased in acute graft rejection patients, while *Firmicutes* decreased. At the family level, *Bacteroides*, *Enterobacteriaceae*, *Streptococcaceae,* and *Bifidobacteriaceae* were increased in acute graft rejection patients, while, *Enterococcaceae*, *Lactobacillaceae*, *Clostridiaceae*, *Ruminococcaceae*, and *Peptostreptococcaceae* were increased in patients without acute graft rejection. Among 38 patients enrolled, 8 suffered from BSI sustained by different Gram-positive (e.g., *Staphylococcus aureus* and *Enterococcus* spp.) and Gram-negative bacteria (namely, *Escherichia coli* and *Pseudomonas aeruginosa*). In conclusion, this study underlines the importance of gut microbiota evaluation for the prevention of complications potentially occurring during the perioperative period.

Another recent prospective, cohort, closed-label study conducted by Salimov UR et al. assessed the impact of gut dysbiosis on acute graft rejection [39]. Twenty-four cirrhotic patients were enrolled, and gut microbiota composition was analyzed from stool samples at different time points (24 h before LT, 1 week after LT, and 1 to 2 weeks after LT) using 16S rRNA NGS. The immunosuppression protocol included the use of basiliximab, tacrolimus, and methylprednisolone in different periods. *Chlorobia* taxa was significantly more abundant (*p* = 0.01) among Child-Pugh Score classes A and B, and the *Coprothermobacterota* taxa (*p* = 0.03) among classes A and C. *Coprothermobacteria* (*p* = 0.01), *Chlorobia* (*p* = 0.04), *Blastocatellia* (*p* = 0.003), *Ktedonobacteria* (*p* = 0.01), *Fimbriimonadia* (*p* = 0.03), *Vicinamibacteria* (*p* = 0.02), and *Chitinophagia* (*p* = 0.02), showed a statistically significant difference between patients with compensated and decompensated LC, respectively. Acute graft rejection differences between groups were found in *Proteobacteria* (*p* = 0.001), *Gammaproteobacteria* (*p* = 0.01); *Chloroflexia* (*p* = 0.004), *Chlamydia* (*p* = 0.01), *Enterobacteriaceae* (*p* = 0.001), and *Candidatus Saccharibacteria* (*p* = 0.04) abundance. The median value of the latter was found to be lower in patients with an acute graft rejection episode. Furthermore, seven patients were stratified into five groups according to infectious complications (e.g., surgical site infections, catheter-associated infections, nosocomial pneumonia, and urinary tract infections). Surgical site infection was the most common complication (21.7% of cases) and was statistically associated with *Aquificae* (*p* = 0.004), *Firmicutes* (*p* = 0.04), and *Fusobacteria* (*p* = 0.03) taxa. As reported by the authors, *Proteobacteria*, *Enterobacteriaceae*, and bacterial strains that have not yet been studied in detail, such as *Candidatus Saccharibacteria*, may play a key role in the development of acute graft rejection. Indeed, more randomized controlled trials to corroborate this hypothesis are necessary.

### 3.4. Gut Microbiota and Dysbiosis Related to Multi-Drug Resistant Bacteria Colonization or Infection

Few studies have evaluated the dysbiosis related to multi-drug resistant (MDR) bacteria colonization or infection in pre- and post-LT patients. Lu H et al. analyzed the gut microbiota composition of patients before and after LT [40]. Twelve patients were prospectively enrolled, providing stool samples at different time points: the last week before the operation and weekly within the first month after the operation. Laboratory analysis was performed with denaturing gradient gel electrophoresis (DGGE) and 16S rRNA gene sequencing using specific primers for V3 conserved regions. The authors showed a substantial reduction in microbial diversity in eight patients during the postoperative follow-up. Among these, five patients with a postoperative hospital stay of more than 30 days showed infections by MDR bacteria (e.g., *Enterococcus faecium* and *Acinetobacter baumannii*). On the opposite, patients with a short postoperative stay did not show significant changes in gut microbial diversity and did not suffer from infections. Thus, the authors underlined the importance of monitoring the gut microbiota composition in patients undergoing LT, as a rearrangement of the eubiosis condition and appropriate prophylactic therapy can prevent the occurrence of infections during the postoperative period.

Annavajhala MK et al. prospectively enrolled 177 adult patients undergoing LT at a single tertiary care center to assess whether colonization by MDR bacteria (e.g., carbapenem-resistant *Enterobacteriaceae*; CRE, *Enterobacteriaceae* resistant to third-generation cephalosporins; Ceph-RE and vancomycin-resistant enterococci; VRE) is associated with intestinal dysbiosis [41]. The 16S V3-V4 rRNA sequencing was performed on 723 fecal samples collected before LT and periodically up to one year after LT to evaluate the microbial diversity. The α-diversity evaluation showed that CRE, VRE, and Ceph-RE colonization were associated with a reduction in both Shannon and Chao α-diversity across all time points, singularly (Shannon *p*  <  0.0001; Chao *p*  <  0.01). Further, VRE-colonized patients showed the most drastic reduction in both diversity indices (Shannon of 2.32  ±  0.86 vs. 3.08  ±  0.73 and Chao of 189  ±  73 vs. 250  ±  75, respectively). At the same time, UniFrac β-diversity showed that patients colonized with MDR bacteria, within one-year post-LT, had also significantly altered β-diversity across all time points (PERMANOVA *p*  =  0.001). As additive evidence, the use of antibiotic therapy resulted in an alteration of microbial diversity (*p* = 0.001), independently of the timing of sample collection. Finally, low pre-LT Shannon α-diversity was associated with subsequent post-LT colonization from any MDR bacteria (*p* < 0.05). Interestingly, pre-LT β-diversity was significantly altered in patients that developed MDR bacteria colonization at any time post-LT vs. patients who were never colonized (PERMANOVA *p* = 0.021). Patients never developing MDR bacteria colonization had a higher pre-LT abundance of *Faecalibacterium prausnitzii*, *Bacteroides* and *Bifidobacterium*, and of *Parabacteroides distasonis*, *Prevotella copri* and *Prevotella stercorea* (*p* < 0.05). The authors concluded the study by emphasizing the importance of gut dysbiosis screening to prevent complications and colonization by MDR bacteria during the post-LT clinical course.

Case-control studies regarding LT are summarized in Table 1.

## 4. Infective Complications after Liver Transplantation

LT is performed to improve life expectancy and quality of life in patients with advanced chronic liver disease. Therefore, continuous monitoring of the patient during the postoperative course is essential to pursue these goals [42]. Bacterial infections are the main complication of LT, often leading to the death of the patient [43]. Increased bacterial infection rates were observed, ranging between 36% and 69% in the post-transplantation period [44]. At any time, some microorganisms can, as a result of gut dysbiosis, migrate to other body districts favoring the establishment of an infectious process. In detail, some bugs present in the hospital setting are MDR, and their colonization is usually associated with increased mortality in liver transplantation candidates [45]. Specifically, Ferstl PG et al. showed how the incidence of MDR bacteria colonization increased during the period before LT. In agreement, MRD bacteria colonization was associated with increased wait-list mortality (Hazard Ratio; HR = 2.57, *p* < 0.0001). Therefore, decolonization strategies should be continuously applied to these patients [46]. In addition, a study conducted by Cardile S. et al. on pediatric LT candidate patients colonized by carbapenemase-resistant *Klebsiella pneumoniae* (CR-KP) suggests that targeted antibiotic therapy is a viable choice when treatment is effective but not prophylactic [47]. MDR bacterial colonization promotes the establishment of infections over time, with a high risk of death for these patients [48].

We know that the most recurrent complications after liver transplantation are surgical site infections (10% of cases), intra-abdominal infections (27–47% of cases), BSIs (13–60% of cases), and hospital-acquired pneumonia (HAP) (5–48% of cases), respectively. These are sustained by different microorganisms, as reported in Figure 2 [49,50,51].

Focusing on the first two, surgical site and intra-abdominal infections are certainly provoked by “direct” damage at the operating site, especially if aseptic surgery is not applied [52]. The role of the gut microbiota in these infections still needs to be better elucidated through clinical studies. However, studies in mouse models have shown the first promising results. In fact, the study by Velasco et al. showed mice undergoing fecal microbiota transplantation being less prone to surgical site infections [53]. On the other hand, probiotics products (e.g., the extracellular polysaccharide EPScG11 of *Lactobacillus plantarum* that reduces the expression of pro-inflammatory factors and increases the expression of anti-inflammatory factors in rats) could be used as a treatment for intra-abdominal infections [54].

Finally, the topic is hot for clinicians because the previously reported infections, promoted by dysbiosis of the gut microbiota, usually occur in the first month post-surgery and are associated with a high risk of liver graft rejection through biochemical pathways that should be elucidated [55].

## 5. Immune System and Gut Microbiota

The gut microbiota is generally localized within the intestine, interacting with the intestinal mucosa. In this district, a homeostatic balance between the microbiota and the intestinal immune response is established. This equilibrium is possible because of the immune tolerance towards the microbiota, finalizing the limitation of intestinal permeability [56]. The balance can be disrupted by different factors, including the development of LC. Cirrhosis is a chronic inflammatory state of the liver of all patients undergoing LT. This pathological condition is characterized by a reduction in synthetic and detoxifying processes and altered metabolic and immunological processes, leading to extensive damage to liver tissue [43,57]. The latter, in turn, causes an increase in permeability of the intestinal mucosa. Subsequently, the microbiotas’ components (e.g., gram-negative polysaccharide) reach the already compromised liver through portal venous circulation and can develop inflammatory processes. Thus, microbial dysbiosis finds its pathogenic expression [33,43,57]. In this section, we will analyze the interactions between microbial dysbiosis and the immune system and how this modulates cirrhosis and subsequent LT.

### 5.1. Gut Microbiota and Liver Cirrhosis

The translocation of microbiota to the liver following cirrhosis triggers the immune response, causing worsening of tissue damage or promoting the onset of further infectious foci [58,59]. Specifically, bacteria activate various PRR receptors in myeloid cells within the liver, resulting in an increased production of Interferon I (IFN), which in turn causes the overproduction of IL-10. Subsequently, the high production of IL-10 results in the downregulation of several inducers of the innate immune response (namely, INF-γ, IL-12, and IL-1β). This ends up in the reduction in phagocytic and bactericidal activity of myeloid cells. Therefore, the progression of inflammation and liver damage can be observed, with the development of fibrosis and, finally, HCC [17]. This IL-10-mediated pathway may be further enhanced by reduced T cell activation, resembling the complexity of immunological pathways. This pathway is reported in Figure 3A.

A second immunological pathway involves CD27 memory B cells. In fact, it has been observed that high exposure of these cells to bacterial lipopolysaccharide (LPS), following microbiota translocation, results in both a quantitative reduction in the cells and dysfunction in those that persist [59]. Dysfunction is mainly a “compromised” activation and production of Tumor Necrosis Factor-beta (TNF-β) and Immunoglobulin G (IgG), and finally, a reduced capability to recruit Th cells [60]. The quantitative reduction in CD27 memory B cells, on the other hand, is thought to be caused by excessive production of Fas. The latter process makes the cells more susceptible to apoptosis (e.g., mediated by the receptor tyrosine kinases Ras) [43]. This pathway is reported in Figure 3B.

### 5.2. Beneficial Effects of Gut Microbiota on Liver Transplantation

The beneficial effects of the microbiota mainly concern a reduction in the inflammatory state and the consequent increase in tolerance of the transplanted organ. These effects occur through various mechanisms that have not yet been fully elucidated. We have reported the mechanisms that are currently accepted. It has been observed that the microbiota can improve tissue damage in the small intestine through the activation of Nucleotide-binding Oligomerization Domain 2 (NOD2) receptors. In particular, microbiotas’ metabolites activate a series of PRRs in intestinal stem cells, which leads to the increased production of Nuclear Factor kappa-light-chain enhancer of activated B cells (NF-kB). NF-kB is a transcription factor for NOD2 that, once expressed, promotes activation of the autophagy process, improving intestinal tissue damage [61]. Further, the microbial metabolite 3,4-dihydroxyphenylpropionic acid (3,4-DHPPA) is obtained from a bacterial reduction in caffeic acid and seems to have anti-inflammatory activity. Presumably, 3,4-DHPPA inhibits Histone Deacetylase (HDAC) activity, leading to a reduction in the pro-inflammatory activity of macrophages and a decrease in the inflammatory state [62]. This pathway is reported in Figure 4A. It has also been hypothesized that increased transplantation tolerance occurs due to a reduction in the activity of effector T cells. This is possible through various mechanisms that are not yet fully clarified. An important role is played by Treg cells that release inhibitory cytokines that interact with CD80/CD86 receptors and inhibit the action of T cells through the Cytotoxic T-Lymphocyte-associated Antigen 4 (CTLA-4) immune checkpoint [63]. In this way, Tregs would help the tolerance of the transplanted organ [58]. Another mechanism recalls the anti-inflammatory activity of IL-10. When there is an increase in exposure to bacterial LPS in the liver tissue following the translocation of the microbiota, high production of IL-10 by numerous immune cells occurs in response. Further, IL-10 induces the activation of the Signal Transducer and Activator of the Transcription 3 (STAT3) receptor in Treg cells, allowing inhibition of T cells [64]. It is also hypothesized that overexposure to gram-negative LPS determines the activation of the CD80 receptor. The latter, through the Programmed Death 1 (PD-L1) immune checkpoint, regulates the apoptosis and activation of effector T cells [65,66]. This pathway is reported in Figure 4B. Although the increase in tolerance of the transplanted organ is a welcome effect, the excessive suppression of the immune system produced by these processes brings the patient more susceptible to further inflammatory processes caused by the translocated microbiota or by MDR bacteria that have colonized the graft site after surgery, increasing the mortality risk [58].

### 5.3. Negative Effects of Gut Microbiota on Liver Transplantation

It is now known that the gut microbiota influences the immune system, determining the outcome of liver transplantation through the modulation of inflammation [67,68]. It has been reported that in post-LT patients, the microbiota was associated with the severity of liver diseases and the presence of acute cellular rejection (ACR) [38]. In light of this, identifying the immunological pathways that correlate the translocation of the gut microbiota to transplant rejection represents a crucial key to improving patient conditions in the post-LT period. Currently, there is limited scientific evidence in the literature directly correlating an immunological pathway triggered by the translocation of the microbiota to liver organ rejection. The crucial point seems to be the disruption of the balance between Tregs and effector T cells caused by the microbiota, resulting in an increase in effector T cells at the expense of Tregs. It has been observed that post-LT ischemia/reperfusion injury leads to excess production of IL-17, which in a murine model has led to a reduction in Tregs and an increase in alloreactive T cells, consequently leading to the development of ACR [69,70]. The following pathway is illustrated in Figure 5. Additionally, other potential immunological pathways have been hypothesized, which are already known to be triggered by the microbiota, and altering the balance of Tregs/effector T cells, could lead to ACR. It has been suggested that the disruption of this balance in mesenteric lymph nodes could result in an increase and migration of CD4+ T cells, promoting liver damage and ACR development [58,71,72]. Dysbiosis is also associated with elevated production of pro-inflammatory cytokines in the host, some of which appear to be directly involved in increasing the risk of ACR onset [58,73,74].

## 6. Conclusions and Future Directions

LT is a treatment applied to patients with severe liver damage, such as cirrhosis and HCC. This event, at the expense of long waiting lists, allows a significant improvement in the patient’s quality of life. However, the chance of organ rejection may result in its failure. The microbiota that populates the gut–liver axis could modulate this response through specific immunological pathways that should be deeper elucidated. Consequently, a therapeutic and preventive approach through the use of prebiotics, probiotics, and symbiotics designed to promote eubiosis should be applied to each patient before surgery. Finally, continuous monitoring of the gut microbiota during the postoperative course is essential to prevent infective complications.

## Figures and Tables

**Figure 1 biomedicines-11-01792-f001:**
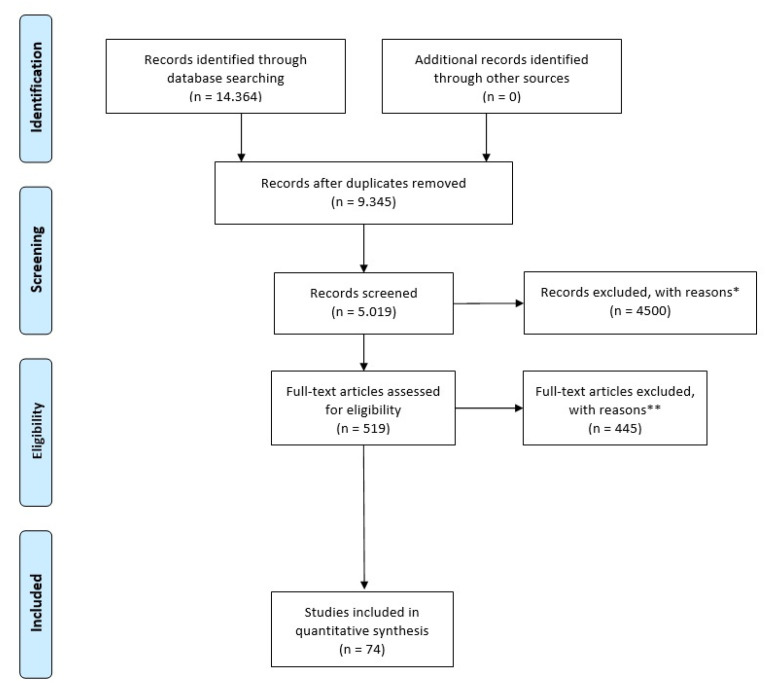
PRISMA flow diagram. * Records were excluded due to the following reasons: (1) duplicated full texts; (2) unavailable full texts; (3) abstract-only papers. ** Full-text articles were excluded due to the following reasons: (1) the articles did not report data for individual comparison groups; (2) case-control studies did not specify the biological sample used.

**Figure 2 biomedicines-11-01792-f002:**
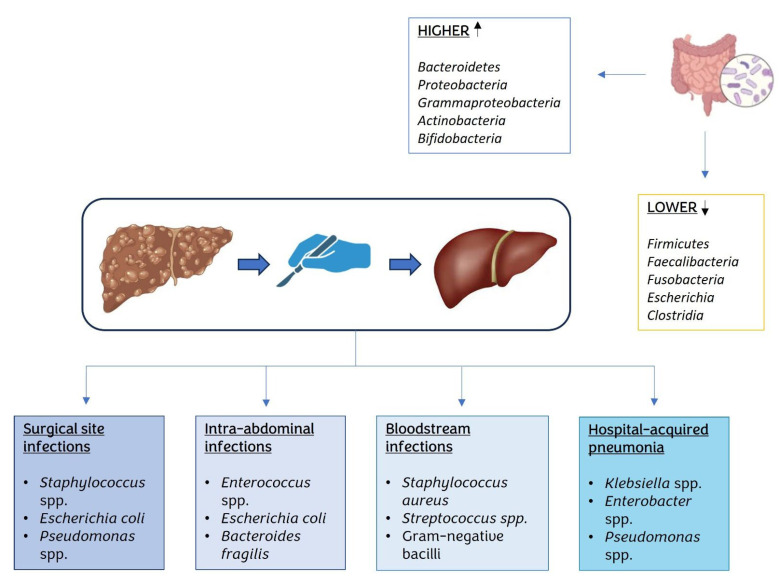
Schematic representation of changes in gut microbiota composition (namely, abundance) and of infective complications after liver transplantation. Black arrow: “up” and “down” indicates the changes in abundance.

**Figure 3 biomedicines-11-01792-f003:**
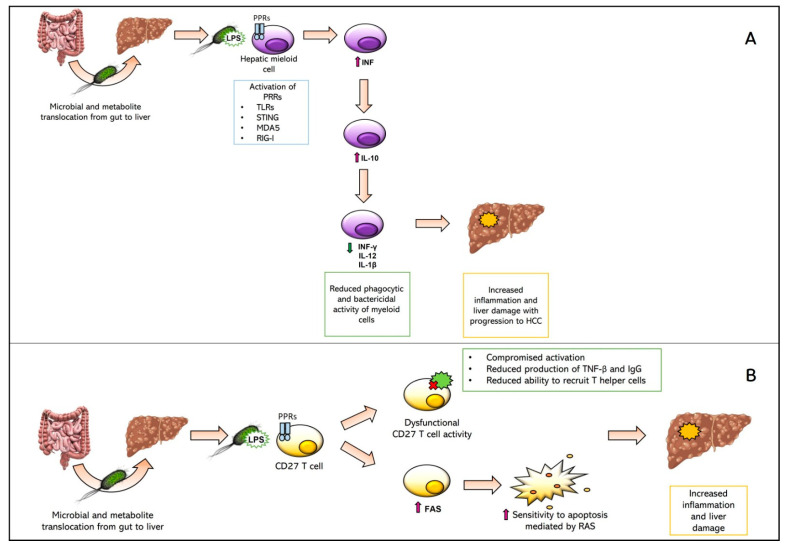
Schematic representation of immunological pathway mediated by INF and IL-10 (Panel **A**) and by CD27 T cells (Panel **B**).

**Figure 4 biomedicines-11-01792-f004:**
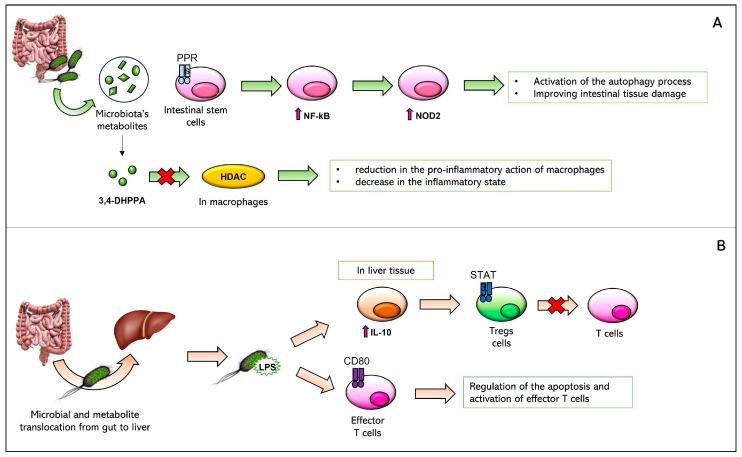
Schematic representation of immunological pathways mediated by gut microbiota metabolites (Panel **A**) and by LPS (Panel **B**).

**Figure 5 biomedicines-11-01792-f005:**
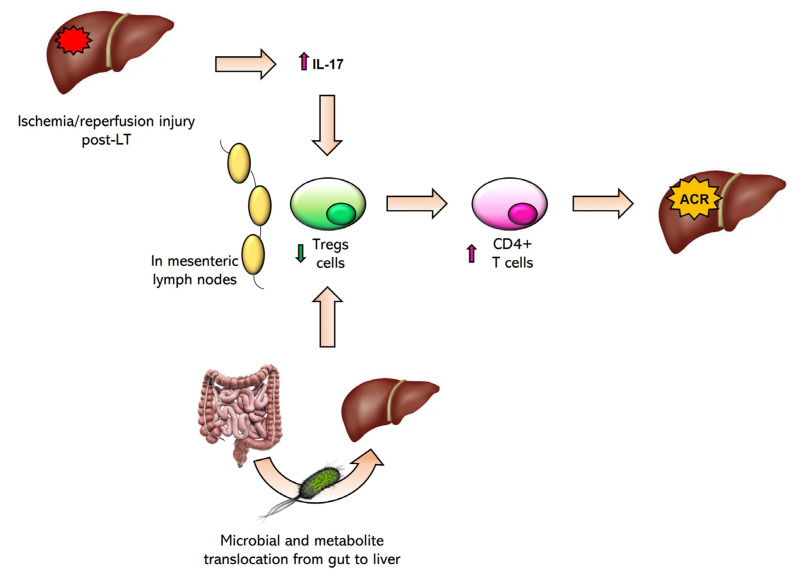
Schematic representation of immunological pathways mediated by the downregulation of Treg cells.

**Table 1 biomedicines-11-01792-t001:** Summary table of case-control study regarding liver transplantation.

Reference	n Patients	Cirrhosis Etiology	LT Period	Sequencing Highlights
Grat M. et al., 2015 [30].	40 patients with LC.	Infective, alcoholic, autoimmune, primary sclerosing cholangitis, Wilson’s disease, non-alcoholic steatohepatitis, primary and secondary biliary cirrhosis.	Pre-LT.	Correlation between PTDR and *Enterococcus* counts; Association between serum bilirubin *Bifidobacterium*, *Enterococcus*, and yeast numbers; Correlation between *Escherichia coli* counts and INR, and *Bifidobacterium* counts with serum creatinine; *Bifidobacterium* and *Enterococcus* counts were independent predictors of the MELD score.
Vangara S. et al., 2022 [31].	14 LT candidates vs. 14 healthy donors.	Alcoholic, dysmetabolic.	Pre-LT.	Increase in *Bacteroidetes* and decrease in *Proteobacteria* in the donor group vs. LT candidates; Correlation between *Proteobacteria* and *Enterobacteriaceae* and increase in MELD score; Association between portal hypertension and increase in *Actinobacteria*, *Bacteroidetes*/*Firmicutes* ratio and decrease in *Lactobacillaceae*, *Bifidobacteria*, and *Ruminococcus* abundance.
Wu ZW et al., 2012 [32].	51 patients with LC vs. LT patients and 28 healthy subjects,	Infective.	Pre- and post-LT.	Lower number of copies of *Faecalibacterium prausnitzii* and *Lactobacillus* spp. in the LT group than in healthy control subjects; Higher number of copies of *Enterobacteriaceae* and *Enterococcus* spp. in the LT group than in healthy control subjects; Lower number of copies of *Faecalibacterium prausnitzii* in liver cancer patients vs. liver cirrhosis patients.
Sun LY et al., 2017 [33].	9 patients with an end-stage liver disease vs. 15 healthy controls.	Infective, alcoholic.	Pre- and post-LT.	Decrease in the relative abundances of *Actinobacillus*, *Escherichia*, *Shigella*, *Fusobacteriales*, *Clostridium*, and *Aeromonas* in post-LT samples compared with pre-LT samples; Increase in the relative abundance of *Desulfobacterales*, *Sarcina*, and *Akkermansia* in post-LT samples compared with pre-LT samples.
Bajaj JS et al., 2017 [34];	68 cirrhotic outpatients: pre-LT group vs. post-LT group vs. controls;	Infective, alcoholic, dysmetabolic.	Pre- and post-LT.	Reduction in relative abundance of potentially pathogenic bacteria and increase in potentially beneficial bacteria.
Bajaj JS et al., 2018 [35].	40 cirrhotic outpatients: pre-LT group vs. post-LT group vs. controls.	Infective, alcoholic, dysmetabolic.		Significant increase in TMAO and PAG in pre- e post- LT; Significant changes in diacylglycerols, cholesterol esters, lyso-phosphethonolamine/phosphoethanolamine, and sphingomyelin after LT.
Lee SK et al., 2022 [36].	27 post-LT patients: 22 long-term post-LT vs. 5 tolerant patients.	Infective.	Post-LT.	Decrease in *Faecalibacterium prausnitzii*, *Bifidobacterium longum,* and *Bifidobacterium bifidum* in post-LT patients vs. control subjects; Decrease in Treg cells in long-term post-LT patients compared to healthy subjects.
Lai Z. et al., 2022 [37]	37 patients at different time points: before LT (BLT; 16 samples), one week after LT (LT1W; 16 samples), and two weeks after LT (LT2W) vs. the control group (CG; 21 samples).	Not specified.	Pre- and post-LT.	Reduction in the α-diversity in LT1W and LT2W groups vs. the other groups; Significant diversity between CG and BLT groups and between BLT and LT1W and LT2W groups (β-diversity); Higher abundance of *Firmicutes* and *Bacteroidetes* in each study group; Higher abundance of *Bacteroides* and *Faecalibacterium* in CG and BLT groups.
Kato K. et al., 2017 [38].	38 LT candidates.	Infective, alcoholic, primary biliary cirrhosis, non-alcoholic steatohepatitis.	Pre- and post-LT.	Increase in *Proteobacteria*, *Actinobacteria, Bacteroides*, *Enterobacteriaceae*, *Streptococcaceae* and *Bifidobacteriaceae* in acute graft rejection patients; Decrease in *Firmicutes* in acute graft rejection patients; Increase in *Enterococcaceae*, *Lactobacillaceae*, *Clostridiaceae*, *Ruminococcaceae*, and *Peptostreptococcaceae* in patients without acute graft rejection.
Salimov UR et al., 2023 [39].	24 patients with LC.	Infective, alcoholic, autoimmune, cryptogenic, Wilson’s disease.	Pre- and post-LT.	*Chlorobi* taxa were significantly more abundant between Child-Pugh Score classes A and B, and the *Coprothermobacterota* taxa among classes A and C; Acute graft rejection differences between groups were found in *Proteobacteria*, *Gammaproteobacteria*, *Chloroflexia*, *Chlamydia*, *Enterobacteriaceae* and *Candidatus Saccharibacteria*; Surgical site of infection was statistically associated with *Aquificae*, *Firmicutes*, and *Fusobacteria* taxa increased abundance.
Lu H. et al., 2013 [40].	12 LT candidates.	Infective.	Pre- and post-LT.	Reduction in microbial diversity during the postoperative follow-up; Infections with MDR bacteria in patients with a postoperative hospital stay of more than 30 days.
Annavajhala MK et al., 2019 [41].	177 patients undergoing LT.	Infective, alcoholic, dysmetabolic.	Pre- and post-LT.	Association between CRE, VRE, and Ceph-RE colonizations and a reduction α-diversity across all time points; Altered β-diversity across all time points in patients colonized with MDR bacteria; Higher pre-LT abundance of *Faecalibacterium prausnitzii*, *Bacteroides*, *Bifidobacterium*, *Parabacteroides distasonis*, *Prevotella copri* and *Prevotella stercorea* in patients who never developed MDR bacteria colonization.

Abbreviations: PTDR: Pre-Transplantation Dysbiosis Ratio; INR: International Normalized Ratio; MELD: Mayo End-Stage Liver Disease; LT: Liver Transplantation; TMAO: trimethylamine-N-oxide; PAG: phenylacetylglutamine; Treg: regulatory T; LT1W: one week after liver transplantation; LT2W: two weeks after liver transplantation; CG: control group; BLT: before liver transplantation; MDR: Multi-Drug Resistant; CRE: Carbapenem-Resistant *Enterobacteriaceae*; VRE: vancomycin-resistant enterococci; Ceph-RE: *Enterobacteriaceae* resistant to third-generation cephalosporins.

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
