# Peer review of "Gut Microbiota and Liver Transplantation: Immune Mechanisms behind the Rejection"

_biomedicines, 2023, doi:10.3390/biomedicines11071792_

Round 1

Reviewer 1 Report

The review is well-done and only minor changes should be made to improve quality and be published in biomedicines.

·       Abstract

·       Material and methods. Please, include the reasons to exclude 4500 records from the 5019 screened such as: Unrelated, duplicated, unavailable full texts, or abstract-only papers.

·       Results and discussion sections. Authors could reorganize the studies included in these sections according to a more simple classification such as:

o   1) gut microbiota and dysbiosis related to prognosis in the waiting list

o   2) gut microbiota and dysbiosis after LT

o   3) gut microbiota and dysbiosis related to graft rejection

o   4) gut microbiota and dysbiosis related to MDRI

·       Section 3.1 and Table 1. This section is very extensive and the information regarding the studies from 27 to  38 should shorten to capture the ideas with clarity, brevity, and precision in short sections.

·       Figures 2 and 3 could be combined into one for a better understanding 

·       Figures Eight Figures are probably excessive and some of them could be included as supplementary information 

·       Section 5. Lines 362-382 could be included in the introduction section

·       Figures 4 and 5 could be combined into one for better comprehension.

Author Response

#Reply to Reviewer 1:

We would like to thank the reviewer and the editorial board for their consideration of our manuscript and their insightful comments. We feel we have responded to the Reviewer’s queries and we hope that now the manuscript will be suitable for publication.

In detail:

  1. Abstract:

Reply 1: Lines 23-25 were added.

  1. Material and methods section: Please, include the reasons to exclude 4500 records from the 5019 screened such as: Unrelated, duplicated, unavailable full texts, or abstract-only papers.

Reply 2: The reasons to exclude 4500 records were added in the text (lines 71-72).

  1. Results and discussion sections: Authors could reorganize the studies included in these sections according to a more simple classification such as:

- gut microbiota and dysbiosis related to prognosis in the waiting list

- gut microbiota and dysbiosis after LT

- gut microbiota and dysbiosis related to graft rejection

- gut microbiota and dysbiosis related to MDRI

Reply 3: The text has been revised and organized in line with the suggestion (see lines 94, 134, 251, 296)

  1. Section 3.1 and Table 1: This section is very extensive and the information regarding the studies from 27 to 38 should shorten to capture the ideas with clarity, brevity, and precision in short sections.

Reply 4: The section has been modified, and Table 1 was revised.

  1. Figures 2 and 3: could be combined into one for a better understanding.

Reply 5: Figures 2 and 3 are now summarized in Figure 2 (line 363).

  1. Figures: Eight Figures are probably excessive and some of them could be included as supplementary information.

Reply 6: Thanks for your comment. We reduced the number of figures to five.

  1. Section 5: Lines 362-382 could be included in the introduction section.

Reply 7: Fixed in lines 52-59.

  1. Figures 4 and 5: could be combined into one for better comprehension.

Reply 6: Both figures were summarized in Figure 3 (line 416).

Reviewer 2 Report

This is a review article on the effect of gut microbiota and liver transplant immune mechanism.

I have some comments.

1.        (L321, 347) “liver transplant” will be “liver transplantation”.

2.        (L384) A schema will be helpful to understand the section.

3.        (L418) The title of section is “negative impact of gut microbiota on liver transplant”. However the reference 56 is the issue on the cirrhosis. In L420, it is described that cirrhosis or LT. Is this section on the graft or cirrhotic liver? Figure 4 indicated that the section is on the cirrhosis not transplanted liver. It is unlikely that the similar phenomenon will happen in both cirrhotic liver and liver graft.

4.        (L413) Is this paragraph about the cirrhosis? The references 40 and 66 are the issues on the cirrhosis.

5.        (L445) Is this paragraph about the post-transplant liver? The references 67-70 are not those on post-transplant liver. The livers in figure 5 are cirrhotic.

I think that the Quality of English Language is good.

Author Response

#Reply to Reviewer 1:

We would like to thank the reviewer and the editorial board for their consideration of our manuscript and their insightful comments. We feel we have responded to the Reviewer’s queries and we hope that now the manuscript will be suitable for publication.

In detail:

  1. (Lines 321, 347) “liver transplant” will be “liver transplantation”.

Reply 1: Fixed in lines 365.

  1. (Line 384) A schema will be helpful to understand the section.

Reply 2: A schema was added as Figure 4 (line 454).

  1. (Line 418) The title of section is “negative impact of gut microbiota on liver transplant”. However the reference 56 is the issue on the cirrhosis. In L420, it is described that cirrhosis or LT. Is this section on the graft or cirrhotic liver? Figure 4 indicated that the section is on the cirrhosis not transplanted liver. It is unlikely that the similar phenomenon will happen in both cirrhotic liver and liver graft.
  2. (Line 413) Is this paragraph about the cirrhosis? The references 40 and 66 are the issues on the cirrhosis.
  3. (Line 445) Is this paragraph about the post-transplant liver? The references 67-70 are not those on post-transplant liver. The livers in figure 5 are cirrhotic.

Replies 3-5: Thank you for your comment. Your suggestions allowed us to observe that paragraph 5.2. needed more clarity regarding pathways related to dysbiosis and cirrhosis/liver transplantation. Therefore, we have divided the immunological pathways into separate paragraphs: from 5.1. to 5.3 (Lines 395, 419, 457). We believe that this may have significantly improved the quality of the manuscript.

Round 2

Reviewer 2 Report

This is a review article on the effect of gut microbiota and liver transplant immune mechanism.

I have some comments.

1.        (L337, Table 1) The abbreviations need explanation at the bottom of the table: PTDR, MELD, LT, TMAO, PAG, Treg, CG, BLT, MDR, CRE, VRE and LT1W.

2.        (L452, Figure 4B) The liver in figure 4B are cirrhotic, which should be replaced by a normal like liver (like a liver right in the blue square in the Figure 2).

3.        (L455) Again, is this section truly on the liver graft? No references support the title of the section (references 67, 68 are not on the transplanted liver). It is described in L458 that “Other immunological mechanisms have also been reported in the literature: they exacerbate the immune response, causing Acute Cellular Rejection (ACR).” However no references on the microbiota and acute cellular rejection are cited here.

4.        (L467) The Figure 4 may be Figure 5.

5.        (L469, Figure 5) The livers in the figure 5 should be replaced by normal livers suggesting a graft if the section 5-3 is truly on the liver graft not on the deceased liver.

I think that the quality of English language is perfect.

Author Response

Reply to Referee #2

Dear Referee, thank you for your new suggestions. We made a second round revision, according to your comments. We hope that this version of our manuscript, is suitable for publication.

Regards

  1. (L337, Table 1) The abbreviations need explanation at the bottom of the table: PTDR, MELD, LT, TMAO, PAG, Treg, CG, BLT, MDR, CRE, VRE and LT1W.

Reply 1: Abbreviations with explanations were added in lines 338-342.

  1. (L452, Figure 4B) The liver in figure 4B are cirrhotic, which should be replaced by a normal like liver (like a liver right in the blue square in the Figure 2).

Reply 2: Fixed.

  1. (L455) Again, is this section truly on the liver graft? No references support the title of the section (references 67, 68 are not on the transplanted liver). It is described in L458 that “Other immunological mechanisms have also been reported in the literature: they exacerbate the immune response, causing Acute Cellular Rejection (ACR).” However no references on the microbiota and acute cellular rejection are cited here.

Reply 3: Dear referee thank you for your suggestion. We revise the paragraph and we improve the text with new sentences and updated references. In particular on this topic we found new articles now discussed and reported in the text (L 474-495).

  1. (L467) The Figure 4 may be Figure 5.

Reply 4: Fixed.

  1. (L469, Figure 5) The livers in the figure 5 should be replaced by normal livers suggesting a graft if the section 5-3 is truly on the liver graft not on the deceased liver.

Reply 5: Fixed.

Round 3

Reviewer 2 Report

The authors' responses are perfect.